# Novel Intragenic and Genomic Variants Highlight the Phenotypic Variability in *HCCS*-Related Disease

**DOI:** 10.3390/genes15121636

**Published:** 2024-12-20

**Authors:** Linda M. Reis, Donald Basel, Pierre Bitoun, David S. Walton, Tom Glaser, Elena V. Semina

**Affiliations:** 1Department of Ophthalmology and Visual Sciences, Medical College of Wisconsin, 8701 Watertown Plank Road, Milwaukee, WI 53226, USA; lreis@mcw.edu; 2Department of Pediatrics and Children’s Research Institute, Medical College of Wisconsin and Children’s Wisconsin, 8701 Watertown Plank Road, Milwaukee, WI 53226, USA; dbasel@mcw.edu; 3Génétique Médicale, SIDVA91/Altérité, 1 Impasse de la Cour de France, 91260 Juvisy-sur-Orge, France; bitoun@gmail.com; 4Department of Ophthalmology, Massachusetts Eye and Ear Infirmary, Harvard Medical School, 8 Hawthorne Place, Boston, MA 02114, USA; walton.blackeye@gmail.com; 5Department of Cell Biology and Human Anatomy, University of California, Davis, CA 95616, USA; tmglaser@ucdavis.edu; 6Department of Cell Biology, Neurobiology and Anatomy, Medical College of Wisconsin, 8701 Watertown Plank Road, Milwaukee, WI 53226, USA

**Keywords:** *HCCS*, microphthalmia with linear skin defects (MLS), corneal opacity, corneal leukoma, Peters anomaly, aphakia, duplication

## Abstract

**Background**: Disruption of *HCCS* results in microphthalmia with linear skin lesions (MLS) characterized by microphthalmia/anophthalmia, corneal opacity, aplastic skin lesions, variable central nervous system and cardiac anomalies, intellectual disability, and poor growth in heterozygous females. Structural variants consisting of chromosomal rearrangements or deletions are the most common variant type, but a small number of intragenic variants have been reported. **Methods**: Exome sequencing identified variants affecting *HCCS*. **Results**: Three novel intragenic variants and two genomic deletions of *HCCS* were found in individuals with primarily ocular features of MLS. X-inactivation was highly skewed in affected individuals with all three intragenic variants. Corneal opacity was the most penetrant feature (100%). In addition, a duplication of uncertain significance including both *HCCS* and *AMELX* was identified in a male with corneal anomalies, glaucoma, an atrial septal defect, and enamel hypoplasia along with a family history of developmental ocular disorders consistent with X-linked inheritance. **Conclusion**: Although variable expressivity is a known feature of MLS, our findings provide additional support for including *HCCS* in testing for individuals with isolated ocular anomalies and provide further evidence for its association with congenital aphakia, aniridia/other iris defects, and corneal staphyloma/ectasia.

## 1. Introduction

Microphthalmia with linear skin defects (MLS; MIM # 309801) is an X-linked dominant disorder mapped to Xp22 in 1990 via individuals with translocations and deletions [1,2]. Holocytochrome c synthase (*HCCS*) was later identified as the causative gene within the region [3]. Major diagnostic criteria are microphthalmia/anophthalmia and irregular linear skin lesions affecting the face and neck at birth, which typically fade over time. Additional clinical findings may include poor growth, developmental delay, and brain, cardiac, gastrointestinal, or genitourinary anomalies, with the latter two more commonly seen in individuals with Xp22 deletions [4,5]. MLS predominantly affects females and is likely to be lethal for males, as the handful of affected males reported have two X chromosomes (46,XX with X/Y translocation) or low-level mosaicism for the disruption [4,5].

HCCS is localized within the mitochondrial inner membrane, where it is responsible for the formation of holocytochrome c by catalyzing the attachment of heme to cytochrome c [6]. The His154 and His211 residues are highly conserved and directly involved in HCCS enzymatic activity [7]. Four conserved domains (I through IV) were identified through the alignment of HCCS homologues [6]. Functional testing through the mutation of residues within each domain identified roles in heme binding (domains I and II), localization to the mitochondria (III), and folding and/or stability of the protein (IV). HCCS function is required for normal apoptosis as well as apoptosome-independent caspase-9-mediated cell death [5]. Variants in two other X-linked genes encoding mitochondrial proteins, *COX7B* and *NDUFB11*, have phenotypes overlapping MLS, but without microphthalmia or corneal abnormalities [5].

Chromosomal/gene deletions (structural variants, SV) affecting Xp22/*HCCS* remain the most common mechanism, with more than 79 individuals reported to date [5,8,9,10,11,12]. Ten different intragenic variants have been reported in eleven families with phenotypes that overlap MLS, including eight pathogenic or likely pathogenic alleles [3,10,13,14,15,16,17,18,19,20,21]. Premature truncating variants cluster in the final two exons of the gene. Three missense alleles affect conserved residues within domains II and IV, and a *de novo* in-frame duplication allele was identified upstream of domain I. Two strong variants of uncertain significance consist of a splice site variant, c.608+5G>A, with *in silico* models strongly predicting disrupted splicing resulting in a premature termination codon, and a late nonsense variant, p.(Trp263*), but without parental testing for either. Skewed X-inactivation is typical in affected females, consistent with a selective growth disadvantage for *HCCS*-deficient somatic cells [5].

The clinical features of *HCCS* deficiency are highly variable, with some genetically diagnosed individuals having only ocular or only skin findings. Indeed, occasionally female carriers are clinically normal [3,19]. This variability is independent of deletion size and thought to arise from differences in X-inactivation patterns in various tissues of affected individuals [5]. Ocular and skin lesions are the most penetrant features, affecting over three-quarters of individuals with SV in one review [5]. The typical ocular phenotype for SV is microphthalmia or anophthalmia (in 77%), with corneal abnormalities in many (64%). Central nervous system malformations of the corpus callosum, septum pellucidum, or ventricles and short stature were noted in about half, while cardiac anomalies, including congenital heart defects, tachycardia, or cardiomyopathy, and intellectual disability were seen in one-quarter to one-third. Among the five individuals with intragenic *HCCS* variants presented in the same review, the distribution of features was similar, with variability likely due to the small size of the cohort [5]. Here, we present additional individuals with *HCCS* variants, including the first reported duplication of this gene.

## 2. Materials and Methods

This study was approved by the Institutional Review Boards of the Medical College of Wisconsin, University of Michigan and University of Iowa, with written informed consent obtained from every participant. Clinical features were provided by referring physicians or medical records. Exome sequencing was performed by Psomagen (Rockville, MD, USA), as previously described [22], and analyzed using VarSeq 2.6.1 software, which includes a CNV caller (VS-CNV; [23]) (Golden Helix, Bozeman, MT, USA). Variants were scored by ACMG/AMP criteria [24,25] to assess pathogenicity. Frequency in the general population was determined in comparison to the gnomAD v4.1.0 database [26].

X-chromosome inactivation (XCI) studies were performed on peripheral blood DNA at Greenwood Genetic Center (Families 1, 3, and 6) by PCR using a polymorphic CAG repeat in the first exon of the Androgen Receptor (AR) gene, as previously reported [27]. In Family 2, X-inactivation studies were performed at UC-Davis using a modified HMGB3 gene PCR assay due to homozygosity for the AR repeat segment in the proband [28], using primers 5′-FAM-CGTGGAGGCAGCTAGCGCGAG and 5′-GCCGCCTAGCAGGAGGGAAGG. To correct for differential amplification, the ratio of allelic products from *Hpa*II-digested genomic DNA (inactive X) was normalized to the ratio obtained from undigested DNA.

To determine the frequency of features in individuals with intragenic variants in *HCCS*, all literature reporting likely/pathogenic or strong variants of uncertain significance (defined as <2 alleles in gnomAD with features consistent with MLS spectrum) was reviewed. The denominator for each feature is the number of individuals with clinical information provided regarding that feature. For growth and intellectual development, only individuals over 6 months of age at the time of the report were counted as a negative case due to the later age of onset for these features.

## 3. Results

Variants in *HCCS* (NM_005333.5), including four intragenic alleles (Figure 1), were identified in 10 individuals from 8 families with corneal opacity (Table 1). Six are new (Figure 2) and presented below (three likely pathogenic intragenic variants, two pathogenic deletions, and one duplication of uncertain significance) and two were previously reported [11,17]. No pathogenic/likely pathogenic alleles were identified in other genes associated with developmental eye disorders in these individuals.

Individual 1 is a 13-year-old female. Ocular features include right microphthalmia, opaque cornea with no view of the iris or posterior pole, and phthisis, as well as left Peters anomaly, absent iris, anterior staphyloma, and entropion with normal eye size. She was adopted at age 4 from China, so her early history is unknown, but no scarring or pigmentation anomalies were present on the face or neck area. She has no other anomalies. Exome sequencing identified a novel c.404G>A p.(Trp135*) variant in exon 5 (of 7) seen in 32/67 reads that was not present in gnomAD. It meets the ACMG criteria for likely pathogenic (LP) (PVS1, PM2_supp) and was confirmed by Sanger sequencing. X-inactivation was highly skewed (100:0 allele ratio). Parental samples are not available.

Individual 2 is a female who was evaluated at 22 months of age with right extreme microphthalmia, thin vascularized cornea with multiple iridocorneal attachments (inferior aspect), and endothelial opacity, as well as a left internal Descemet’s membrane defect similar to Peters anomaly, with a superior medial iris defect and normal eye size. No additional clinical features were reported at the time of enrollment (age 7). Exome sequencing identified a novel c.603G>A p.(Trp201*) variant in exon 6 (of 7) seen in 89/197 reads that was not present in gnomAD. The variant meets the ACMG criteria for LP (PVS1, PM2_supp), was confirmed by Sanger sequencing, and was not present in the unaffected mother’s sample; the unaffected father was not available for testing. X-inactivation was highly skewed in the proband (2:98), with inactivation of the maternal allele, and balanced in the mother (59:41).

Individual 3 is a female who presented in infancy with bilateral microphthalmia, Peters anomaly, and sclerocornea. The non-ocular findings included short stature and left ventricular anomaly with short PR interval progressing to atrioventricular block at age 15. Exome sequencing identified a novel c.650G>A p.(Arg217His) variant in 34/65 reads that was predicted to be damaging by CADD, REVEL, and AlphaMissense (with scores of 28.2, 0.775, and 0.856, respectively) and was ultra-rare in gnomAD (1/1210474 alleles; 0 hemizygotes). The variant meets the ACMG criteria for likely pathogenic (PM2_supp, PM5, PP3_mod, PP4) and was absent in the unaffected mother (0/93 reads); the unaffected father was not available for testing. X-inactivation was highly skewed (96:4) in individual 3 with inactivation of the maternal allele; the mother’s sample was moderately skewed (90:10). Clinical re-evaluation of individual 3 at age 31 after the genetic diagnosis identified a single linear defect scar under her chin, normal intelligence, and a reported history of Wolf–Parkinson–White syndrome, which was successfully treated by catheter ablation.

Individual 4A is a female who was evaluated at 4 months of age with bilateral corneal opacity. In the right eye, the pupil and clear lens were visible through the opacity, while the left eye showed an extreme anterior segment defect with absent anterior chamber. Her mother, individual 4B, has right Peters anomaly (status post-enucleation) and a left superior medial sector iris defect. No skin or dental anomalies were reported. Copy number analysis of exome data identified an at least 3.49 Mb heterozygous deletion of X:8095006-11682950 (hg19), including *VCX2*, *VCX3B*, *ANOS1*, *FAM9A*, *FAM9B*, *TBL1X*, *GPR143*, *SHROOM2*, *CLDN34*, *WWC3*, *CLCN4*, *MID1*, *HCCS*, *ARHGAP6*, and *AMELX* in the child and mother. Within this region, all 11 informative variants with read depth >10 were homozygous in the proband. Since there is sufficient evidence for *HCCS* haploinsufficiency in the ClinGen database [29], the deletion meets the ACMG pathogenic criteria.

Individual 5 is a 14-month-old female with unilateral ocular anomalies consisting of left primary congenital aphakia with corneal opacity, absent iris, and microphthalmia, and a normal right eye. Non-ocular anomalies include neonatal hypoglycemia and preauricular pits. No information is available regarding her dental findings. Copy number analysis of exome data identified a ~239 kb heterozygous deletion of X:11130179-11369520 (hg19), including *AMELX*, *ARHGAP6*, and *HCCS* (exons 2-7 only), that was not present in either unaffected parent. This *de novo* deletion was confirmed by independent clinical copy number analysis of exome data in the proband and meets the ACMG pathogenic criteria.

Individual 6A is a male who presented with bilateral corneal edema and opacity at 3 weeks of age. Exam under anesthesia at 4 weeks noted normal intraocular pressure (IOP) and was diagnosed as having either a primary corneal pathology or congenital glaucoma with nanophthalmos; treatment with dorzolamide and latanoprost eye drops was initiated. Both corneas were edematous and hazy but without Haabs striae, and the iris and lens were normal in both eyes. Ocular hypertension was noted at 2 months of age and timolol drops were added. At 4 months of age, IOP was further elevated with increasing axial length and worsened cup to disc ratio (0.6, 0.65), prompting Harms trabeculotomies in both eyes. Eye drops were stopped after surgery but increasing IOP in the right eye necessitated resuming drops within a week. Slight buphthalmos and ptosis were noted at 9 months with mild myopia. Non-ocular anomalies include mild hypertelorism, laryngopharyngeal reflux, laryngomalacia, redundant duodenum, a history of failure to thrive, large-volume vomiting in infancy requiring NG (nasogastric) tube feeding (resolved by 10 months of age), torticollis, sacral indentation, and eczema. Dental examination revealed generalized enamel decalcification, erosion (maxillary anterior, possibly due to reflux/vomiting), and severe early childhood caries, with all molars requiring extraction or crowns at age 4. Echocardiogram revealed a small ostium secundum atrial septal defect with left-to-right shunt, persistent left superior vena cava (normal variant), and aortopulmonary collateral connection. The pedigree reveals a history of congenital cataracts and/or glaucoma in two of his five brothers and seven unaffected sisters. In the extended family, there are two maternal great uncles, one of whom also had congenital cataract and glaucoma, and the maternal grandmother has a diagnosis of adult-onset glaucoma and reported loss of night vision in her 40s. The proband’s mother (individual 6B) is currently 44 years old with hyperopia. Copy number analysis of the exome data identified a ~189 kb duplication X:11130179-11318734 (hg19), including *AMELX*, *ARHGAP6*, and *HCCS*, which was hemizygous in the proband and heterozygous in his mother. X-inactivation studies showed a balanced XCI ratio in the mother (54:46). The duplication was confirmed by clinical laboratory qPCR in the proband. No other family members were available for testing. In the ClinGen database of dosage sensitivity, *HCCS* is rated as having ‘no evidence’ for triplosensitivity (last evaluated 10 May 2012) [29], so the duplication is considered a variant of uncertain significance (VUS). Since *HCCS* is on the edge of duplication, it is possible that position effects decrease *HCCS* expression, but the random X-inactivation pattern in the mother argues against this since haploinsufficiency results in highly skewed inactivation patterns. If the overexpression of *HCCS* is the cause of the phenotype, it may be that random X-inactivation lessens the effects in females by lowering the overall dose.

Two additional individuals with *HCCS* disruption were reported by us previously. Patient 3 in Weh et al. (2014) [17] with a c.715C>T p.(Gln239*) variant had an isolated ocular phenotype with bilateral Peters anomaly and mild microphthalmia. Primary aphakia was diagnosed in her right eye while an abnormal residual lens was present in the left. Individual 9 in Reis et al. (2023) [11] with a ~4.08 Mb deletion including *HCCS* had a clinical diagnosis of Axenfeld–Rieger syndrome with bilateral congenital glaucoma, polycoria, microphthalmia (right smaller than left), corneal opacity with failed corneal transplant in adulthood, and small congenital cataracts. Non-ocular anomalies included small, mispositioned teeth, enamel hypoplasia, and extra teeth, but no other congenital/developmental anomalies.

Overall, the rate of non-ocular features of MLS was lower in our cohort compared to previous reports. Only one individual with a pathogenic/likely pathogenic variant had additional anomalies (cardiac, growth, and skin), and the skin lesion consisted of a single subtle linear scar that was identified only after genetic diagnosis; no CNS or cognitive anomalies were identified. Due to the substantial increase in the number of intragenic alleles reported since the last review, we compiled an updated summary of the features for intragenic alleles (Table 2). The rates of CNS malformations, intellectual disabilities, and short stature in individuals with intragenic variants decreased, but all other rates remained similar to the initial cohort of five individuals. Notably, 100% of individuals with intragenic variants and clinical details reported have corneal opacity and 92% have micro/anophthalmia. Overall, the prevalence of various features of MLS (ocular, CNS, skin, intellectual disabilities, and short stature) is similar among cases with SV or intragenic variants, consistent with a predicted loss-of-function mechanism for intragenic variants.

## 4. Discussion

The likely/pathogenic *HCCS* variants presented here are primarily associated with ocular features. While variable expressivity is a known feature of MLS, the rate of associated non-ocular features in our cohort is lower than previously reported [5]. This most likely reflects an ascertainment bias. Because linear skin lesions are a key diagnostic feature of MLS, clinical testing is more likely to be undertaken on individuals meeting multiple MLS criteria. Conversely, individuals with non-syndromic ocular anomalies are less likely to undergo clinical testing and more likely to enroll in a congenital eye disease research study. In contrast to prior reviews, corneal opacity was the most penetrant ocular feature in our cohort (100%); microphthalmia was noted in 75% of cases but was often mild. Peters anomaly, characterized by central corneal opacity (CCO) with iridocorneal and/or keratolenticular adhesions and posterior corneal defects, was a common clinical diagnosis, and Descemet’s membrane defects were specifically noted in two cases. Consistent with the recent characterization of corneal phenotypes in other individuals with MLS [10], corneal staphyloma and/or corneal ectasia were noted in two individuals in our cohort. Iris sector or circumferential defects and/or congenital aphakia were also observed in five individuals; while not typically considered features of MLS, both aniridia and congenital aphakia have been noted in other cohorts [10,12,30]. In many cases, the degree of corneal opacity precludes an examination of the anterior segment, so the prevalence of these features is likely to be higher. The absence of the lens during early development can cause corneal opacity, microphthalmia, and iris defects [31], suggesting a possible mechanism for the panocular features of MLS. Other known causes of congenital primary aphakia include recessive loss-of-function variants in *FOXE3* and *PXDN* and specific dominant variants in *GJA8* [31], as well as *Pitx3* deficiency in mice [32].

Our findings increase the number of reported intragenic variants to 13 (in 14 individuals). The novel missense variant c.650G>A p.(Arg217His) affects the same amino acid as the previously reported p.(Arg217Cys) allele. In HCCS functional assays, Cys, Glu, Lys, and Ala substitutions of Arg217 each eliminated enzyme activity, likely due to protein folding/stability effects, suggesting that a His substitution is also unlikely to be tolerated [6]. With regard to the premature truncation variants (PTV), the majority of alleles introduce an erroneous stop codon within the final 50–55 nucleotides of the penultimate exon (exon 6) or in the final exon and are thus expected to escape mRNA nonsense-mediated decay (NMD) (Figure 1). Of note, while the previously reported frameshift allele occurs upstream of this cut-off, the 30 amino acid erroneous tail places the premature stop codon within the final exon. However, most of the truncating alleles occur upstream of the critical active sites of the protein, prior to or within a mitochondrial targeting signal, and all but one disrupt or eliminate the conserved domain IV, shown to be critical for protein folding and/or stability [6]; thus, the truncated protein products, if produced, are expected to be non-functional. One newly identified nonsense allele, p.(Trp135*), occurs in exon 5 and is thus expected to trigger NMD. Only one reported nonsense allele, p.(Trp263*), does not affect any conserved domains and is considered a variant of uncertain significance.

All three deletions presented here include *AMELX* in addition to *HCCS.* Loss of *AMELX* [33] is known to cause amelogenesis imperfecta, type 1E (MIM #301200). One individual displayed the typical dental features and was initially given a diagnosis of Axenfeld–Rieger syndrome due to the combination of characteristic ocular and dental defects [11]; no information is available regarding dentition in the other three individuals with deletions. Given the proximity of *AMELX* and *HCCS*, individuals with MLS and genomic deletions should be referred for dental evaluation if *AMELX* is involved.

The effect of the *HCCS* duplication is unknown. The presence of a corneal phenotype in the proband along with a family history consistent with X-linked inheritance suggests that it may be causative, via increased *HCCS* dosage or a position effect on gene expression. It is likely that several of the unaffected sisters are also carriers of the duplication and may develop a later-onset phenotype similar to the grandmother. X-inactivation studies in the mother did not show skewing in the blood, making loss of function due to a position effect less likely. Interestingly, the male proband had enamel anomalies and severe childhood caries, suggesting that the duplication of *AMELX* may also result in dental problems. In the DECIPHER database [34], there are 17 males with duplications <30 Mb that include *HCCS*. Clinical information is available on ten; of these, two were noted to have glaucoma with abnormal dental enamel in one. Further clinical data from males with *HCCS* duplications are needed to determine the clinical significance of this variant.

This study expands the genotypic spectrum for *HCCS* with novel missense and nonsense variants and the first report of *HCCS* duplication. Our findings suggest that it is important to screen *HCCS* in individuals with isolated ocular anomalies and provide further evidence for its previously reported association [10] with congenital aphakia, aniridia/other iris defects, and corneal staphyloma/ectasia.

## Figures and Tables

**Figure 1 genes-15-01636-f001:**
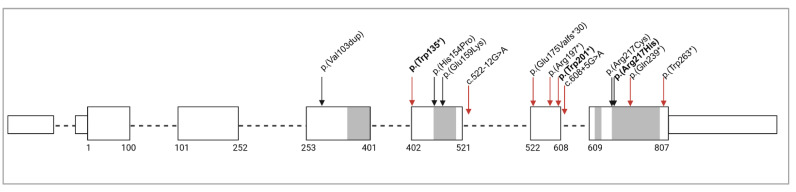
Intragenic variants in *HCCS.* Schematic of *HCCS* with location of variants identified in individuals with features consistent with MLS. Exons are indicated with boxes (non-coding regions are shorter boxes) and introns with a dashed line. Coding sequence positions are marked below each exon. Gene regions encoding conserved domains I through IV are indicated in gray. Black arrows indicate missense and red arrows indicate premature truncation variants. Variants in bold are novel variants presented in this study.

**Figure 2 genes-15-01636-f002:**
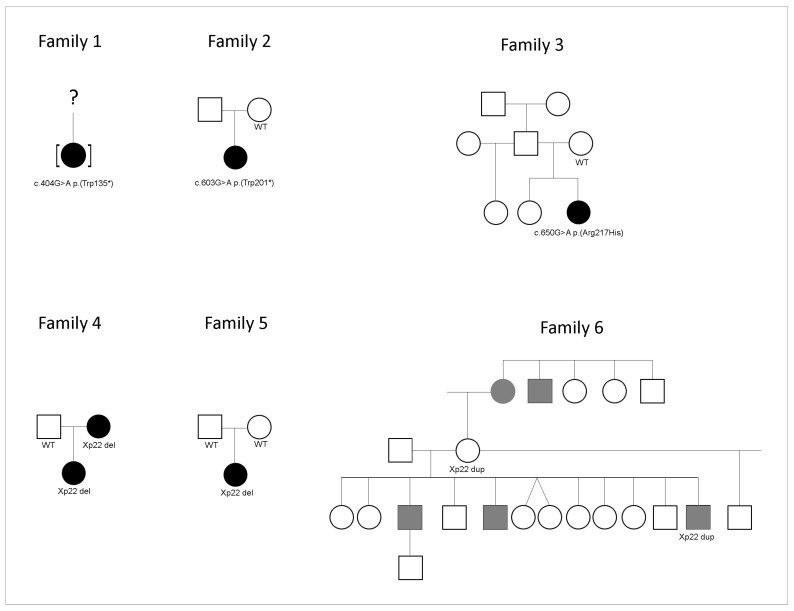
Family pedigrees for individuals with *HCCS* variants. Solid black symbols indicate ocular features of MLS; solid gray symbols indicate other ocular conditions (see text). Specific *HCCS* variants or WT (wild type) are indicated under tested individuals in each family.

**Table 1 genes-15-01636-t001:** Summary of genotypic and phenotypic information of individuals in this study.

Ind	Genomic Position ^a^	Variant ^b^	ACMG ^c^	GnomAD (v4.1.0)	XCI Ratio	Age, Gender	Skin	Eye	CNS	ID	SS	Heart
1	X:11136623	c.404G>A p.(Trp135*)	LP (PVS1, PM2_supp)	NP	100:1	13 y, F	NR	+	-	-	NR	-
2	X:11139108	c.603G>A p.(Trp201*)	LP (PVS1, PM2_supp)	NP	2:98	7 y, F	NR	+	NR	NR	NR	NR
3	X:11139773	c.650G>A p.(Arg217His)	LP (PM2_ supp, PM5, PP3_mod, PP4)	1/1210474 (0 hemi)	96:4	31 y, F	+	+	-	-	+	+
[17] (#3)	X:11121718	c.715C>T p.(Gln239*)	LP (PVS1, PM2_supp)	NP	-	5 m, F	-	+	-	-	-	-
4A	X:8169987-11664819	~3.49 Mb deletion: *VCX2*, *VCX3B*, *ANOS1*, *FAM9A*, *FAM9B*, *TBL1X*, *GPR143*, *SHROOM2*, *CLDN34*, *WWC3*, *CLCN4*, *MID1*, *HCCS*, *ARHGAP6*, *AMELX*	P (2A, 4A)	NP	-	4 m, F	NR	+	NR	NR	NR	NR
4B	X:8169987-11664819	~3.49 Mb deletion: *VCX2*, *VCX3B*, *ANOS1*, *FAM9A*, *FAM9B*, *TBL1X*, *GPR143*, *SHROOM2*, *CLDN34*, *WWC3*, *CLCN4*, *MID1*, *HCCS*, *ARHGAP6*, *AMELX*	P (2A, 4A)	NP	-	adult, F	NR	+	NR	NR	NR	NR
5	X:11130179-11369520	~239 kb deletion: *HCCS*, *ARHGAP6*, *AMELX*	P (2A, 4A, 5A)	NP	-	14 m, F	-	+	-	-	NR	-
[11] (#9)	X:7370404-11445756	~4.08 Mb deletion: *VCX*, *PNPLA4*, *VCX2*, *VCX3B*, *ANOS1*, *FAM9A*, *FAM9B*, *TBL1X*, *GPR143*, *SHROOM2*, *CLDN34*, *WWC3*, *CLCN4*, *MID1*, *HCCS*, *ARHGAP6*, *AMELX*	P (2A, 4A, 5A)	NP	-	57 y, F	-	+	-	-	-	-
6A	X:11130179-11318734	~189 kb duplication: *HCCS*, *ARHGAP6*, *AMELX*	VUS	NP	-	5 y, M	-	+	-	-	-	+
6B	X:11130179-11318734	~189 kb duplication: *HCCS*, *ARHGAP6*, *AMELX*	VUS	NP	54:46	44 y, F	-	-	-	-	-	-

^a^ hg19 coordinates. ^b^ Intragenic based on NM_005333.5, gene lists based on HGNC protein coding genes within the region. ^c^ Variant interpretation following ACMG guidelines [24,25]. Clinical features: ye, microphthalmia, and/or corneal opacity; CNS, central nervous system malformation; ID, intellectual disability; SS, short stature. Abbreviations: LP, likely pathogenic; P, pathogenic; VUS, variant of uncertain significance; NP, not present; m, months; y, years; F, female; M, male; +, present; -, absent; NR, not reported.

**Table 2 genes-15-01636-t002:** Comparison of clinical features in affected individuals presenting with intragenic *HCCS* alleles.

Skin Lesions	Micro/Anophthalmia	Corneal Opacity	CNS Malformations	Developmental Delay	Short Stature	Cardiac Anomalies	References ^a^
4/9 44%	12/13 92%	11/11 100%	4/12 33%	2/7 29%	2/8 25%	7/13 54%	[3,10,13,14,15,16,17,18,19,20,21]

^a^ One variant reported in three papers [13,14,15] represents the same individual and is counted once here.

## Data Availability

All variants were submitted to ClinVar. There are no other data associated with this manuscript.

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
