# Peer review of "Novel Intragenic and Genomic Variants Highlight the Phenotypic Variability in HCCS-Related Disease"

_genes, 2024, doi:10.3390/genes15121636_

Round 1

Reviewer 1 Report

Comments and Suggestions for Authors

The research article titled "Novel variants including the first gene duplication and phenotypic variability in HCCS-related disease" by Ries et al. is well-written, and presents its findings in an organized manner. I believe, this study represents a valuable contribution to the medical literature, offering important insights into HCCS-related conditions. Specific comments and suggestions have been provided directly in the manuscript to enhance its clarity and impact.

Author Response

Reviewer 1:

The research article titled "Novel variants including the first gene duplication and phenotypic variability in HCCS-related disease" by Ries et al. is well-written, and presents its findings in an organized manner. I believe, this study represents a valuable contribution to the medical literature, offering important insights into HCCS-related conditions. Specific comments and suggestions have been provided directly in the manuscript to enhance its clarity and impact.

Response: Thank you for your careful reading of the manuscript. Specific comments and responses are added below:

  • Please add the age of onset for the irregular skin lesions, and for other features (if applicable)
    • Age of onset for the skin lesions was added
  • Please provide the prevalence rate of MLS worldwide and/or for specific cohorts. Also include if any specific ethnicity or geographic clustering of MLS was observed in the current or previous studies.
    • Information about the prevalence of MLS is provided in paragraph 3. It is ultra-rare with only 89 individuals reported in the literature. Since almost all cases are de novo, there is not specific ethnicity or geographic clustering.
  • If possible, do mention/cite the specific CNV caller that detected deletions and a duplication reported in this study. Also, comment on efficiency of CNV detections by exome sequencing vs genome with reference to this particular study
    • The CNV caller is part of the VarSeq software. The methods were modified to make this clearer. A reference supporting the effectiveness of the VarSeq CNV caller has been added.
  • Please improve the formatting of this table (Table 1).
    • The formatting was corrected.
  • Include exon number/total exons for pLoF variants.
    • This information was added
  • How PVS1 is applicable to a missense variant? And why was PP3 not applied if the in-silico predictions were damaging? Please clarify the ACMG class for this variant.
    • We apologize for the error. The ACMG criteria for the missense variant (as noted in Table 1) are PM2_supp, PM5, PP3_mod, and PP4, still meeting the LP class. This has been corrected in the text.
  • Though genome assembly is given in the footnotes of the table 1, it maybe a good idea to include it with the genomic variant position in the text as well.
    • This was added
  • Happloinsufficiency
    • This was changed
  • The carrier mother had 07 daughters and none of them is affected. Please discuss (perhaps in discussion) the likelihood of the daughters being unaffected carriers.
    • Since the only reported phenotype in females is adult-onset glaucoma, it is very likely that several of the sisters are carriers- no phenotype would be expected at younger ages. More discussion of the siblings was added to the text.
  • Is this also pointing towards incomplete penetrance or this is a sole function of X inactivation which is unlikley considering the XCI in carrier mother.
    • It is possible that inactivating the overexpression in approximately half of cells, as occurs with random X-inactivation in females, is sufficient to protect tissues from the increased dose effect. (This sentence was also added to the text)
  • Is this also not because of the ascertainment bias? Due to this bias, comparison with previous studies may not be a good idea. These stats can still be reported as a finding of this study without any comparisons.
    • This study was open to individuals with any type of developmental ocular anomaly, with or without corneal opacity, so we do not believe that the high rate of corneal phenotypes is a result of ascertainment bias.
  • Could this cause dominant negative effect?
    • The available data suggests loss of function for all truncated proteins due to loss of Domain IV (for all but) and additional disruption of the mitochondrial targeting signal and Domain III for most. There is no evidence to suggest dominant negative function. Additional text was added to the discussion.
  • Is the phenotype of the patient with HCCS duplication not different from a typical MLS? Should this case be subjected to genome sequencing to exclude other genes?
    • We agree that genomic sequencing is the next step to fully evaluate this case and plan to complete this in the future.
  • Again, a small cohort with a possible ascertainment bias - This conclusion may be misleading.
    • We updated this phrase from “Our findings suggest that HCCS-related isolated ocular anomalies may be more common than previously recognized…” to “Our findings suggest that it is important to screen HCCS in individuals with isolated ocular anomalies”

Reviewer 2 Report

Comments and Suggestions for Authors

Microphtalmia with Linear Skin defects is a X-linked dominant disease, predominantly affecting females, associated with skewed X-chromosome inactivation, and likely lethal in male individuals.

Holocytochrome c synthase (HCCS) has been identified as the causative gene of the MLS. The HCCS gene encodes for a mitochondrial protein, localized within the mitochondrial inner membrane. The enzyme functions by catalyzing the bonding of the heme group to the cytochrome c, thus leading to a mature holcytochrome c. Structurally the HCCS protein is organized in four distinct domains. While domains I and II are responsible for the heme-binding, domain III enables the mitochondrial localization of the enzyme. Eventually, domain IV stabilizes the protein.

While structural variants, namely chromosomal/gene deletions, affecting XP22/HCCS are the most frequent abnormalities, intragenic variants are less frequent.

In the manuscript titled "Novel variants including the first gene duplication and phenotypic variability in HCCS-related disease", Reis L.M. and colleagues report three novel intragenic variants and two genomic deletions in females with primarily ocular features of MLS. Remarkably, all the individuals displayed a highly skewed X-chromosome inactivation. Additionally, a duplication was identified in a male patient for the first time.

Overall, the inquiry expands the genotypic spectrum for HCCS.

I have no major concerns, but just a curiosity concerning the missense mutations. Did you ever try to predict the effects of missense mutations with one of the currently available online tools (e.g. PROVEAN, SIFT,…)?

A couple of typos are scattered throughout the main text (e.g., lines 96-97, font size), and they require to be amended.

Author Response

Reviewer 2:

Microphthalmia with Linear Skin defects is a X-linked dominant disease, predominantly affecting females, associated with skewed X-chromosome inactivation, and likely lethal in male individuals.

Holocytochrome c synthase (HCCS) has been identified as the causative gene of the MLS. The HCCS gene encodes for a mitochondrial protein, localized within the mitochondrial inner membrane. The enzyme functions by catalyzing the bonding of the heme group to the cytochrome c, thus leading to a mature holcytochrome c. Structurally the HCCS protein is organized in four distinct domains. While domains I and II are responsible for the heme-binding, domain III enables the mitochondrial localization of the enzyme. Eventually, domain IV stabilizes the protein.

While structural variants, namely chromosomal/gene deletions, affecting XP22/HCCS are the most frequent abnormalities, intragenic variants are less frequent.

In the manuscript titled "Novel variants including the first gene duplication and phenotypic variability in HCCS-related disease", Reis L.M. and colleagues report three novel intragenic variants and two genomic deletions in females with primarily ocular features of MLS. Remarkably, all the individuals displayed a highly skewed X-chromosome inactivation. Additionally, a duplication was identified in a male patient for the first time.

Overall, the inquiry expands the genotypic spectrum for HCCS.

Response: Thank you!

I have no major concerns, but just a curiosity concerning the missense mutations. Did you ever try to predict the effects of missense mutations with one of the currently available online tools (e.g. PROVEAN, SIFT,…)?

Response: In silico predictions for the missense variant identified in Individual 3 are provided in the text. In addition to CADD and REVEL which were already provided, we added the AlphaMissense score, which also predicts the variant to be damaging.

A couple of typos are scattered throughout the main text (e.g., lines 96-97, font size), and they require to be amended.

Response: The manuscript was thoroughly reviewed for typos. The font size issue in lines 96-97 is an artifact introduced from formatting into PDF format by the journal- we will carefully review the final manuscript to ensure accuracy.